# Epidemiology and Prevention of Early Infections by Multi-Drug-Resistant Organisms in Adults Undergoing Liver Transplant: A Narrative Review

**DOI:** 10.3390/microorganisms11061606

**Published:** 2023-06-17

**Authors:** Giovanni Dolci, Giulia Jole Burastero, Francesca Paglia, Adriana Cervo, Marianna Meschiari, Giovanni Guaraldi, Johanna Chester, Cristina Mussini, Erica Franceschini

**Affiliations:** 1Infectious Diseases Unit, Azienda Ospedaliero-Universitaria of Modena, 41126 Modena, Italy; giovanni.dolci90@gmail.com (G.D.); g.burastero@gmail.com (G.J.B.); adriana.cervo@gmail.com (A.C.); mariannameschiari1209@gmail.com (M.M.); 2Infectious Diseases Unit, University of Modena and Reggio Emilia, 41121 Modena, Italy; 238241@studenti.unimore.it (F.P.); giovanni.guaraldi@unimore.it (G.G.); crimuss@unimore.it (C.M.); 3Department of Dermatology, University of Modena and Reggio Emilia, 41121 Modena, Italy; johanna.chester@gmail.com

**Keywords:** liver transplant, multi-drug-resistant organisms (MDRO), antibiotic prophylaxis, post liver transplant infections, preventive measures

## Abstract

Invasive bacterial infections are a leading cause of morbidity and mortality after liver transplant (LT), especially during the first months after LT, and infections due to multi-drug-resistant organisms (MDRO) are increasing in this setting. Most of the infections in patients in intensive care unit arise from the endogenous microflora and, for this reason, pre-LT MDRO rectal colonization is a risk factor for developing MDRO infections in the post-LT. Moreover, the transplanted liver may carry an increased risk of MDRO infections due to organ transportation and preservation, to donor intensive care unit stay and previous antibiotic exposure. To date, little evidence is available about how MDRO pre-LT colonization in donors and recipients should address LT preventive and antibiotic prophylactic strategies, in order to reduce MDRO infections in the post-LT period. The present review provided an extensive overview of the recent literature on these topics, with the aim to offer a comprehensive insight about the epidemiology of MDRO colonization and infections in adult LT recipients, donor-derived MDRO infections, possible surveillance, and prophylactic strategies to reduce post-LT MDRO infections.

## 1. Introduction

Invasive bacterial infections are a leading cause of morbidity and mortality after liver transplant (LT) [1,2], with approximately two third of early deaths due to sepsis [3,4]. Bacterial infections are the most common cause of complications within the first month after LT [2] and are often due to nosocomial organisms [5]. Changes in transplant policies and techniques increased the number of LT, including marginal organs and both donors and LT candidates in critical conditions. These policies increase the risk of time in intensive care units (ICU), leading to increased risk of multi-drug-resistant organism (MDRO) colonization and infections for both liver donors and recipients [6,7].

LT recipients have multiple risk factors for MDRO colonization and infections, including immunosuppression, cirrhosis-related immune-dysfunction, prolonged hospitalization and ICU stay, and previous antibiotic exposure. Patients with end-stage liver disease and those hospitalized in ICU are fragile and at high risk of MDRO colonization. Rectal colonization with MDRO in the pre-LT period is a commonly considered risk factor for developing a MDRO-infection in the post-LT period [8,9,10,11] and it is associated with both increased pre- and post- operative LT mortality [12,13,14], as well as prolonged hospital stay and post-LT surgical complications [14]. It is well known that the majority of infections in any patient recovered in ICU arises from the endogenous microflora of the oropharynx and the gut, whereas cirrhotic and LT patients are at increased risk of bacterial translocation from the gastro-intestinal tract to the liver and the abdominal cavity [15,16]. Further, MDRO infections are frequently not responsive to empiric antibiotic therapy and have limited therapeutic options [17].

In this review, we describe the epidemiology of MDRO colonization and infections in adult LT candidates and recipients, donor-derived MDRO infections, and possible surveillance and prophylactic strategies to reduce post-LT MDRO infections. We performed a narrative review to summarize prevalence, risk factors, and prevention strategies for MDRO colonization and infection in LT candidates/recipients.

## 2. Methods

We reported this review according to the Scale of Assessment of Narrative Review Articles (SANRA) guidelines [18].

### 2.1. Criteria for Considering Studies for This Review

Studies were included if they met the following criteria: (i) prospective, retrospective cohort studies, or case reports; (ii) reported prevalence rates, risk factors, or prevention strategies for MDRO colonization and infection in solid organ transplant, with at least a portion of adult LT candidates/recipients (iii) published after 2000. We excluded studies including pediatric patients.

### 2.2. Identification of the Studies

We performed a literature search on the Medline electronic database from 2000 to January 2023. Reference lists of relevant studies were screened for any further publications. The following search terms “extended-spectrum beta-lactamase producing Enterobacteriaceae” (ESBLE), “carbapenem-resistant Enterobacteriaceae” (CRE), “multi-drug resistant *Pseudomonas aeruginosa*” (MDR-PA), “carbapenem-resistant *Acinetobacter baumannii*” (CRAB), “vancomycin-resistant Enterococci” (VRE), “methicillin-resistant *Staphylococcus aureus*” (MRSA), “liver transplant”, and “solid organ transplant” were used. Duplicate articles were removed from the search results manually.

### 2.3. Studies Selection and Data Collection

One reviewer examined the search results, screened the titles, abstracts and reference lists of identified articles, and evaluated individual study’s eligibility. Full texts of the selected articles were obtained and examined for inclusion criteria. The second author checked full texts for inclusion in case of doubt.

Authors extracted and collected data from studies regarding: (i) study (i.e., authors, year of publication); (ii) country where the study was performed; (iii) pre- and post-LT MDRO colonization; (iv) post-LT MDRO infection, (v) post-LT surgical site infections (SSI) and blood stream infections (BSI). The second author checked the data collected.

### 2.4. Definitions

MDRO was defined according to the 2012 international consensus [19], as bacteria non-susceptible to at least one agent in three or more antibiotic classes [20]. The more recent “difficult-to-treat” definition [21] was not used, since the majority of literature cited in this review used the 2012 international consensus definitions.

## 3. Results

### 3.1. Epidemiology of MDRO Colonization and Infections

BSIs and SSIs were the two most common bacterial infections following LT [22,23,24]. Data regarding BSIs varied widely between different geographical areas and years. BSI-related mortality in the post-LT varied between 13 and 24% [25,26], while septic shock-related mortality was greater than 50% [27]. BSIs in LT recipients were most often associated with intra-abdominal or biliary sources with Gram-negative bacteria, which were the predominant pathogens [24,25,27,28,29].

More recent studies [22,23,30,31] reported SSIs as the most common bacterial infection after LT. Avkan-Oguz et al. [22] reported SSIs in 25.6% of infections observed within the first month after LT, whereas Garcia-Prado et al. [30] reported SSIs in 33.5% of infected patients (with 30.5% of deep/organ-space SSIs). Hellinger et al. [32] identified 18% of SSIs, 13.0% of which were deep/organ-space, during the first post-LT month.

Respiratory infections were reported as the most prevalent post-LT infection in some other studies [33], with a high morbidity and mortality burden, especially in patients with prolonged ICU stay and those requiring orotracheal intubation. One of the main causes of pneumonia was MRSA and MDR *Enterobacteriaceae* infection. In the case of ventilator-associated pneumonia, MDR non-fermentative Gram-negative bacilli were most commonly reported. This was because they are saprophytic and environmental pathogens that can colonize ICUs and respiratory devices [34,35].

Finally, urinary tract nosocomial infections were highly prevalent in LT recipients, especially in patients with urinary catheter or with simultaneous kidney transplant.

During the first few decades of LT, SSIs and BSIs were mostly caused by Gram-negative bacteria [27]. In the mid-1990s, Gram-positive bacteria, particularly *S. aureus* [36,37] and VRE [38], emerged as increasingly common pathogens in the post-LT period. This was probably due to technical surgical advances promoting a decrease in Gram-negative bacteria associated abdominal and SSI. Despite ongoing evolution of transplantation practices, Gram-negative bacilli re-emerged as the predominant bacteria in BSIs and SSIs after LT, with a notable increment in MDR Gram-negative bacilli [24,30,39]. This increment was mostly due to the emergence of ESBLE. ESBLE is a major concern as one-year survival post-LT rates were lower than patients with non-MDRO infections [40,41,42].

The incidence of MDRO colonizations and infections are difficult to estimate, as many studies may have overestimated rates if conducted during an outbreak or underestimate rates if microbiological surveillance was not performed routinely.

Post-LT MDRO infection risk factors can be MDRO specific or shared. Pre-LT exposure to broad spectrum antibiotics, prolonged cold ischemia, increased blood transfusion need during LT, and prolonged endotracheal intubation (>2 h) were among the risk factors shared by different MDROs [14,43,44]. Table 1 summarizes the different types of MDRO and their associated risk factors, pre-LT colonization prevalence, post-LT infection prevalence, and mortality reported in specific studies.

### 3.2. Extended Spectrum β-Lactamase-Producing Enterobacteriaceae

ESBL are a heterogeneous family of enzymes that hydrolyze β-lactamic antibiotics, including extended-spectrum penicillins, aztreonam, and third generation cephalosporins. ESBL-producing organisms usually maintain in vitro susceptibility to the cephamycins (i.e., cefoxitin, cefmetazole, and cefotetan), carbapenems, and β-lactamase inhibitors (e.g., clavulanic acid, sulbactam, and tazobactam) [70]. The most common ESBL was the CTX m β-lactamase [71,72].

Although rates of ESBLE infections in LT recipients varied in different geographical areas and hospitals, they were the most prevalent MDR-Gram-negative bacilli isolates in LT recipients [39,70,73] and their prevalence is growing worldwide [39,74]. According to the literature, almost half of ESBLE-colonized LT recipients developed an ESBLE-infection, with roughly a 12 times higher risk that was 12 times higher compared to non-ESBLE-colonized LT recipients [47].

Independent predictors of ESBLE infection were pre-transplant *Klebsiella pneumoniae* fecal carriage, model for end stage liver disease (MELD) score > 25, preoperative spontaneous bacterial peritonitis prophylaxis, and antimicrobial exposure during the previous month [45]. Table 2 summarizes the epidemiology of ESBLE-colonization and infections in LT-recipient.

Half of the studies in the literature dedicated to ESBLE colonization and infection in LT were from the same French cohort, representing almost 75% of the published LT recipients [45,46,75]. The retrospective study by Bert et al. [75] included 710 patients with pre-LT ESBLE screening and LT between 2001 and 2010. Among them, 29 (4.1%) were ESBLE carriers, mostly *Escherichia coli* (21/29). Over the study period, ESBLE fecal carriage increased from 0% to 10.6%. ESBLE infection during the first 4 months after LT was recorded in 39 (5.5%) of the entire population. However, according to pre-LT ESBLE colonization, infection rates were significantly higher among carriers compared to non-carriers, 44.8% vs. 3.8% (*p* < 0.0001), with risk of ESBLE infection calculated as 11.74 higher among ESBLE-colonized patients. ESBLE infection was significantly associated with in-hospital mortality (28.2 vs. 15.9%; *p* = 0.045). Subsequent studies confirmed these results [45,47,77,78].

Other studies described either ESBLE pre-LT colonization or ESBLE post-LT infections only. O’Connell et al. [41] reported a 21.9% pre-LT ESBLE colonization in a tertiary center in Ireland, without any data on post-LT infections. Kim et al. [76] described 112 BSIs (64 patients) in LT patients, of which 27 were ESBLE BSIs.

In our cohort, between 2010 and 2020, of 473 LT-recipients, 45 (9.5%) were colonized by ESBLE pre-LT. At our center, a target prophylaxis was administered to ESBLE carriers. Post-LT infection (SSI and BSI only) during the first month post-LT was identified only in five patients (1%) [79,80].

### 3.3. Carbapenem-Resistant Enterobacteriales (CRE)

CRE is related to multiple mechanisms, including carbapenemase production, efflux pumps hyperexpression, and porin inactivation. Carbapenemases are a heterogeneous group of β-lactamases that hydrolyze carbapenems. They include:-Ambler Class A serine β-lactamase: e.g., *K. pneumoniae* carbapenemase (KPC);-Ambler Class B metallo-β-lactamases (MBLs): e.g., imipenemase (IMP), Verona integron-encoded metallo-β-lactamase (VIM), and New Delhi metallo-β-lactamase (NDM);-Ambler Class D serine β-lactamases: e.g., including the oxacillinase (OXA)-48-family of enzymes.

The geographical distribution of carbapenemases were different. KPC enzyme was the most common in the United States of America (USA), South America, Southern Europe, Israel, and China [81,82,83]. NDM was the most common in India, Pakistan, and the United Kingdom [84,85]. OXA-48-family was common in North Africa, Turkey, and India [86]. From the initial identification of CRE around 15 years ago, CRE increased worldwide. Due to the clinical impact of numerous hospital outbreaks, especially in high complexity and ICU settings, CRE colonization and infections represented one of the most investigated subjects in transplant patients over the last decade. Most of the studies were performed in Italy and Brazil.

Independent risk factors for post-LT CRE infection included CRE colonization, higher MELD at LT (≥25), higher intraoperative blood loss (>1500 mL), prolonged post-LT ICU stay and intubation, post-LT hemodialysis, combined transplant, biliary complications, reintervention, and rejection [8,9,14,48,49,50,51,52,53,54,55,56]. Moreover, post-LT CRE-infections were associated with mortality rates as high as 70% [48,49,56,87]. Epidemiology studies of CRE colonization and infections in LT-recipient are summarized in Table 3.

CRE colonization rates varied widely, even in the same geographical regions. A retrospective study conducted between 2010 and 2018 in a single center in Brazil reported rates of pre-LT CRE colonization of 12.9%, with a decreasing trend during the study period (23.3 to 6.7%) [8]. Lower rates of pre- LT CRE-colonization were reported in cohort studies conducted in Brazil and Italy [90] over the same study period, [52] and in Italy only (between 2014–2015), reporting rates of 7.5% and 2.5%, respectively. Studies from other countries [48,49,50,51,56,88,89], although reporting different colonization prevalence, confirmed that CRE-colonization is an independent risk factor for CRE-infection development and it is associated with higher post-LT mortality.

On the basis of these findings, researchers elaborated several algorithms for colonization, infection, and mortality prediction. A Brazilian study described a host of predisposing factors for pre-LT CRE colonization, including patient antibiotic exposure, hepato-renal syndrome, worst CLIF-SOFA score [91], and the use of beta-lactam/beta-lactamase inhibitors within 90 days prior to LT. Researchers engaged a machine learning approach to develop a CRE colonization at LT predictive model algorithm [90].

CRE colonization, acquired pre-LT, at LT or immediately after LT, was identified as risk factor for CRE infection. An Italian monocentric cohort study found that between 2010 and 2013 [10], 20 out of 237 (8.4%) LT recipients developed a carbapenem-resistant *K. pneumoniae* (CR-KP) infection within 6 months post-LT (18 BSIs and 2 pneumonia); 11 patients were CR-KP colonized at LT and 30 acquired CR-KP colonization post-LT. Acquired CR-KP colonization post-LT had a highest risk of CR-KP infection (46.7%) compared to patients with CR-KP colonization at LT. Multivariate analysis identified CR-KP rectal carriage at any time, together with post-LT renal replacement therapy, mechanical ventilation >48 h, and HCV recurrence as independent risk factors for infection.

A more recent, large, multicenter, international cohort study [92] reported that among 840 CRE carriers, 29.8% developed a CRE infection within 30 days post-LT; distributed as BSI (36.5%), SSI (26%), and lower respiratory tract infections (23.6%).

Giannella et al. [9] reported that CRE colonization was the strongest risk factor for CRE infection, along with combined transplant, higher MELD at LT, prolonged mechanical ventilation, re-intervention, and rejection. On the basis of these predisposing factors, a normogram to predict 30- and 60-day CRE infection risk was developed [93]. The risk prediction model considered pre- and post-LT colonization, multisite post-LT colonization, prolonged mechanical ventilation, acute renal injury, and surgical reintervention.

A mortality risk score to predict 30-day mortality (INCREMENT-SOT-CPE [94]) was developed and validated [92]. The INCREMENT-SOT-CPE [8] score takes in consideration INCREMENT-CPE mortality score ≥8, no source control, inappropriate empirical therapy, cytomegalovirus disease, and lymphopenia.

### 3.4. Multi-Drug-Resistant Pseudomonas Aeruginosa (MDR-PA)

*Pseudomonas aeruginosa* has a wide variety of mechanisms of antimicrobial resistance related to both inherent chromosomal mutations and transmissible resistance determinants.

The most common resistance mechanisms are:-β-lactam: e.g., efflux pumps, constitutive hyperproduction of chromosomal AmpC, and inactivation of the OprD porin, which contributes to imipenem resistance [95,96];-Fluoroquinolones: mutations in DNA gyrase, topoisomerase, and overexpression of efflux pumps [95];-Aminoglycosides: e.g., expression of efflux pumps, decreased outer membrane permeability, amino acid substitutions in ribosomal proteins, and methylation of 16S ribosomal RNA [96].

MDR pseudomonal phenotypes likely arise from a combination of several resistance determinants [96].

Epidemiology of MDR-PA infections in solid organ transplant (SOT) varied widely between geographical regions and transplant centers. However, the overall, worldwide trend showed an increased MDR *P. aeruginosa* incidence [39,70]. MDR *P. aeruginosa* bacteremia were associated with mortality rates approaching 40% in LT recipients [59]. Risk factors for *P. aeruginosa* BSIs were reported in general SOT studies, without any dedicated to LT recipients. Risk factors in SOT recipients include hospital-acquired BSI and ICU admission within 1 year [11,59]. The epidemiology of MDR-PA colonization and infections in LT-recipients are summarized in Table 4.

Few studies reported MDR-PA colonization and infections, representing a general knowledge gap. A Japanese cohort [97] of 170 living-donor LT reported five infections (2.9%) by MDR-PA during 3 months post-LT, while in a Chinese cohort [34], DR-PA was isolated in only 1 of the 178 isolates of pneumonia during 6 months post-LT.

### 3.5. Carbapenem-Resistant Acinetobacter baumanii (CRAB)

*Acinetobacter baumannii* is a saprophytic lactose non-fermentative Gram-negative bacillus and an opportunistic pathogen. It is often carbapenem-resistant and its most common mechanism of carbapenem resistance is the presence of the class D β-lactamases, including genes OXA-23, OXA-24/40, and OXA-58. Other mechanisms are the reduction in carbapenem-associated outer membrane porin (CarO), associated with higher imipenem minimum inhibiting concentrations, and upregulated efflux pumps [70]. A combination of groin cutaneous swabs and rectal swabs seems to be more sensitive in detecting *A. baumannii* colonization than rectal swab alone [44].

CRAB is extremely difficult to eradicate from healthcare facilities, frequently causing nosocomial outbreaks [44,60,61,62]. For this reason, its epidemiology deeply varied not only between different geographical regions, but also among hospitals in the same areas and wards in the same hospital. *A. baumannii* is often MDR, especially in SOT recipients [11]. Shi et al. [57] reported that 62% of the *A. baumannii* isolated from blood cultures in their cohort was MDR. Therefore, post-LT CRAB infections have high mortality rates of 50–65% [44,60,61,62].

Risk factors for post-LT CRAB infection include pre-LT CRAB colonization, fulminant hepatitis, higher pre-transplant MELD, severe encephalopathy, lower donor body mass index, longer cold ischemia time, prolonged post-LT ICU stay and intubation, post-LT dialysis, and reintervention. Epidemiology studies of CRAB colonization and infections in LT-recipient is summarized in Table 5.

A Brazilian cohort study conducted between 2009 and 2011 reported a CRAB colonization prevalence as high as 11% in LT candidates, the highest MDR-colonization rate among their patients [44,98]. CRAB colonization prevalence raised to 43% in the post-LT period, with 29% recipients developing a CRAB infection.

A Chinese cohort [34] reported 21 cases of CRAB (11.8%) among 178 post-LT pneumonia isolates. Another cohort study conducted in China by Min et al. [60] reported a 2.8% prevalence of CRAB bacteriaemia within 30 days post-LT, with cumulative mortality incidence on days 5, 10, and 30 from the index positive blood culture date, of 58.6%, 65.5%, and 65.5%. Independent risk factors for 30-day CRAB-bacteriaemia included pre-transplant MELD, severe encephalopathy, lower donor body mass index, and reintervention.

In a South Korean cohort [62], 3.6% of LT recipients developed a post-LT CRAB BSI, and reported a 50% associated mortality rate.

### 3.6. Other Non-Fermentative Gram Negative Bacilli

*Stenotrophomonas maltophilia*, *Burkholderia cepacia* complex (BCC), and *Achromobacter xylosoxidans* are less common causes of infections compared to other non-fermentative Gram-negative bacilli but are pathogens of concern in SOT recipients [99]. They are ubiquitous in the environment, specifically water and soil, and are known to cause severe infections, especially in people with underlying pulmonary conditions and in immunocompromised hosts [100,101,102]. When isolated in respiratory samples, it can be difficult to understand when they are simply colonizing or true pathogens. These microorganisms are often difficult to treat due to a variety of intrinsic and acquired resistance traits, ranging from intrinsically resistance to aminoglycosides and β-lactams, including the carbapenems (due to the chromosomally encoded β- lactamases [L1, a MBL, and L2, a serine cephalosporinase] typical of *S. maltophilia* [103,104]), through to high production of efflux pumps, changes in lipopolysaccharide structure, and decreased outer membrane permeability (more characteristic of BCC) [105,106].

As these pathogens are often isolated in underling pulmonary conditions, among SOT, they are most often isolated in lung transplants patients. However, a few case reports were recently reported in LT [99].

A total of 26 *S. maltophilia* BSIs were reported in a recent study [99]. Six (37%) of them occurred in LT patients and, in particular, four occurred within 30 days after LT. No data about eventual pre-LT colonization were available.

More studies are necessary to identify specific risk factor for colonization and infection by these pathogens in LT patients.

### 3.7. Vancomycin-Resistant Enterococcus (VRE)

VRE were reported as the second most frequently occurring MDR pathogen causing nosocomial infections in the USA [107]. The majority of VRE isolates are *Enterococcus faecium*.

LT recipients are the most commonly VRE-colonized patients among SOTs. Risk factors for VRE colonization in LT recipients include hospitalization, paracentesis, previous endoscopic retrograde cholangio-pancreatography, and anti-anaerobic antibiotic therapy. VRE post-LT colonization was associated with increased all-cause 90-day mortality [108], but specific information regarding VRE-infection-related mortality was scarce.

VRE infections are usually preceded by VRE colonization, with a 6.7 times higher risk of infection for VRE-carriers [63,109]. As VRE are part of the intestinal microbiota, they mostly cause SSI or organ/space infections in LT recipients, including biliary tract infections and intra-abdominal abscesses, often associated with BSIs [110].

Cohort studies mostly describe increased morbidity and mortality in VRE-colonized LT-recipients. A 2014 meta-analysis by Ziakas et al. [63] indicated a pooled prevalence for VRE colonization before and after SOT of 11.9% and 16.2%, respectively. LT-specific pooled analysis found a 16% post-LT VRE colonization prevalence [63]. However, the current prevalence is probably higher than these estimates, as trends suggest that over the last decade, VRE-colonization increased both in the general hospitalized population and in LT recipients.

Epidemiology of VRE-colonization and infections in LT-recipient is summarized in Table 6.

A recent retrospective cohort study by Chiang et al. [66] showed that among 343 LT recipients treated between 2014 and 2017, 68 (20%) had pre-LT VRE colonization and, among the remaining 275 LT recipients, 20 (9.8%) acquired VRE post-transplant. Six (2%) patients developed an invasive VRE infection, five in the VRE-colonized group and one in the non-VRE-colonized group (5.7% vs. 0.4%). Mortality at 2 years was 13% in VRE-colonized versus 7% in non-colonized recipients (*p* = 0.085).

A study conducted in the USA [67] found a rate of 27% pre-LT VRE colonization. Post-LT VRE infection developed in 5% of recipients pre-LT VRE colonized and no infections developed in the non-colonized group. A study conducted in South Korea [65] reported a rate of 23% colonized by VRE pre-LT and 20% who acquired VRE post-LT. Post-LT VRE infection developed in 9% patients and the authors recorded a higher risk of mortality in LT recipients who acquired VRE post-LT.

In our cohort, 39 (10.9%) LT recipients treated between 2010 and 2020 were pre-LT VRE colonized [112]. After receiving a VRE target prophylaxis, four (10.2%) pre-LT VRE colonized patients developed a VRE infection (three BSIs and one SSI) within 30 days post-LT. Among non-VRE pre-LT colonized patients, only one patient (0.3%) developed a BSI. However, a cohort from Iran [111] did not find an association between VRE-colonization and increased mortality or longer hospital stay.

### 3.8. Methicillin-Resistant Staphylococcus aureus (MRSA)

The incidence of early post-LT infection by MRSA, once a major cause of early post-LT infections [36,37,69], decreased during the last two decades [39]. The main risk factor for post-LT MRSA infection is pre- or post-LT MRSA nasal colonization [69], with an increased risk of 15.6 times compared to non-colonized recipients (odds ratio 15.6; 95% CI 6.6–36.9) [113]. However, common infection control procedures decreased MRSA colonization in the LT setting [114]. Other risk factors for MRSA infection are prolonged ICU stay and/or invasive mechanical ventilation and cytomegalovirus primary infection [37,115,116]. As MRSA usual colonization sites are nose and skin, in LT recipients, it most commonly causes lung and BSIs, including catheter-related BSIs. Post-LT MRSA infection survival varies in different settings and depending on the site of infection, ranging from 6% in catheter-related bacteremia to as high as 60% in complicated MRSA bacteremia and septic shock [37,68]. Epidemiological studies of MRSA-colonization and infections in LT-recipient are summarized in Table 7.

A Chinese cohort [76] described 13 MRSA BSIs in 222 (6%) LT recipients. The South Korean cohort study by Kim et al. [65] reported a 8.4% (12/142) pre-LT MRSA colonization, with nine (6.3%) recipients acquiring MRSA colonization post-LT. Post-LT MRSA infection was registered in 19 LT recipients, with pneumonia being the most represented clinical manifestation. Pre-LT MRSA-colonized patients had the highest risk of post-LT MRSA infection and those who acquired MRSA post-LT had a higher risk of mortality.

In a Japanese cohort [117] of living donor LT recipients, 13% were colonized by MRSA and 5.7% of LT patients developed a MRSA BSI.

### 3.9. Donor-Derived MDR Bacterial Infection

Bacterial donor-derived infections (DDI) represent one third of donors transmitting infections in SOT, with 65% of them due to Gram-negative bacteria [118]. In this setting, the transmission of MDRO from donor to recipient is a known risk [119], with growing interest due to the global increment of MDRO-colonization and the more frequent procurement of marginal livers to increase access to this procedure. Whilst LT recipients with non-MDRO infected donors do not usually have serious infective complications if a specific antimicrobial treatment is provided [120], the use of livers from MDRO-infected donors can have worse outcomes [121]. In the case of MDRO infection, therapeutic options are limited, absent, or have diminished efficacy. Furthermore, particularly virulent bacteria that are also prone to become MDR, such as *S. aureus* and *P. aeruginosa*, can lead more frequently to invasive infections threatening the survival of the graft and the recipient [122,123,124,125]. Special attention should be paid to potential MDRO organ infection from geographical regions with high prevalence of MDRO colonization. The risk of MDRO donor colonization/infection increased with the duration of terminal donor hospitalization, with 20% of donors MDRO colonized by day 10, and 33% of donors MDRO colonized by day 15 [7]. Risk factors for MDRO donor colonization/infections include hepatitis C viremia, dialysis, prior hematopoietic cell transplant, and exposure to antibiotics with a narrow Gram-negative spectrum [7].

Colonization by MDRO is not a contraindication for organ procurement, according to the *European Guide to the Quality and Safety of Organ for Fransplantation* [126], as long as the colonized tissue remains sealed from the rest of the body and particularly from the transplanted organ. The guidelines from the *American Society of Transplantation Infectious Diseases Community of Practice* are more cautious; although MDRO colonization is not a considered an absolute contraindication for organ procurement, a careful discussion for risk-benefit assessment and an outlined plan for peri- transplant antibiotics are required [127].

Guidelines recommendations are reflected in a reduction in the number of organs transplanted per donor and in a higher match run at which organs are accepted in donors with MDR Gram-negative bacteria [6].

The literature on MDRO DDI is sparse, also because the rate of donor-to-recipient transmission is low. The few data available suggest that 8–15% of LT recipients of MDRO-colonized or infected donors develop a MDRO-DDI [128,129], even if underreporting of cases is highly likely [130].

A retrospective Italian cohort study [128], involving 30 SOT recipients of organs with a carbapenem-resistant Gram-negative bacteria infection or colonization between 2012 and 2013, reported three cases of SOT recipient infection (10%), of whom two were LT recipients and one CR-KP colonization. In the first case, the LT recipient was a 22-year-old girl who developed an intra-abdominal CR-KP- purulent collection, which was successfully treated with surgical drainage and a 7-day course of colistin and tigecycline. The second case was a 50-year-old man who developed an abdominal CR-KP SSI, which was resolved with peri-hepatic drainage.

In a large Chinese cohort study [129] of 724 SOT, of whom 240 were LT, studied between 2015 and 2020, 68 donors had MDRO infections. DDI was registered from 10 (14.7%) donors, transmitted to 22 recipients, 4 (18.1%) of whom died. Among the 22 infected recipients, 6 were LT recipients, 1 died, while the other 5 survived with normal graft function. The causing organism of these six DDIs were two *VRE*, one *CRAB*, one *Enterobacter aerogenes*, and one CR-KP. Unfortunately, this study did not use uniform SOT DDI definitions.

Another USA cohort showed that among 182 patients who underwent LT in 2015 and 2016 [121], 22 had an MDRO-positive donor and the increment in the hazard of post-LT infections associated with MDRO on donor culture was 3.8 (95% CI 0.98–14.43), but only one probable MDRO DDI was detected.

Concerning MRSA, we found only single case studies reported in the literature [122]. Three case reports [122,124] described MRSA transmission from a donor with infective endocarditis to a LT recipient, despite a documented pre-LT bacteremia clearance. Finally, Obed et al. [125] described a living donor-derived Panton–Valentine Leucocidine-positive MRSA transmission that most likely was the causative agent of a hemorrhagic necrotizing pneumonia that was fatal to a LT-recipient. Epidemiological studies of DDI in LT-recipient are summarized in Table 8.

### 3.10. Post-LT MDR Bacterial Infection Prevention

The most effective way to tackle MDRO infections is to reduce and/or prevent MDRO colonization. General operator infection control measures, such as reinforcement of hand hygiene and alcoholic hand solution monitoring, associated with antimicrobial stewardship, are fundamental to reduce the incidence of all the MDRO for patients undergoing LT, especially during ICU stay. Even though few LT-specific studies are available for MDRO-infection control, for the purposes of this review, we generalized data from studies on SOT recipients or inpatients admitted to ICU. Surveillance cultures and carrier isolation were proven to be effective for CRE and MRSA [131], while their efficacy and cost-effectiveness are still controversial for ESBLE and VRE [132], especially in high prevalence settings. Water and environmental decontamination are particularly important for non-fermentative saprophytic bacteria, including *P. aeruginosa* and CRAB [133,134,135,136].

Antimicrobial prophylaxis is administered at transplantation to reduce SSI incidence. The most common causative pathogens are skin and intestinal flora. There is no widely accepted antibiotic prophylaxis for LT, as there is not enough evidence to prefer a single antibiotic regimen. When selecting antibiotics, it is necessary to evaluate local bacterial prevalence, resistance patterns, and patient and donor characteristics. The most commonly used and accepted LT prophylaxis regimens include third-generation cephalosporin plus ampicillin, piperacillin-tazobactam, or amoxicillin-clavulanate [137]. Similarly, optimal antibacterial prophylaxis duration is not well defined, but recently, 24 h prophylaxis was proven to be non-inferior to seventy-two-hour prophylaxis in a randomized-controlled trial [138].

Furthermore, to minimize the impact of donor-transmitted bacteria following SOT, prompt inter-facility communication, antibiotic prophylaxis based on in vitro susceptibility testing, and careful infection control practices are essential [127]. MDRO bacteraemic donors should receive effective antimicrobial therapy for 24–48 h prior to procurement and recipients of these organs should receive a 7-to-14-day course of antibiotics targeting the donor isolate [70,127,139].

#### 3.10.1. ESBL-Producing Enterobacteriacaeae

Contact precautions for ESBLE colonization and/or infection are recommended in non-endemic areas, while the role of these measures in endemic regions is still debated [70]. Active surveillance for ESBLE among asymptomatic SOT before surgery is considered good practice, but no data are published regarding how many transplant centers follow this recent recommendation [140].

Intestinal decolonization is not recommended [141]. Studies reported the development of tobramycin- and colistin-resistance by Gram-negative bacilli after attempted intestinal decolonization with orally administered colistin [142,143].

The role of perioperative prophylaxis targeted on ESBLE-colonization remains uncertain. The 2019 guidelines from the American Society of Transplantation Infectious Diseases Community of Practice [70] did not recommended target prophylaxis.

Nevertheless, after these guidelines, a paper by Logre et al. [45] reported a higher incidence of ESBLE infections after LT among ESBLE carriers comparing inactive versus active ESBLE prophylaxis. Of the 11 patients who received inactive ESBLE prophylaxis, 63.6% developed an infection, compared to 29.8% of the 57 patients who received an active ESBLE prophylaxis. Although this difference was statistically significant (29.8% vs. 63.6%, *p* = 0.04), the study had several limitations: its retrospective nature, the small sample size, and the imbalance between the two prophylaxis groups. Despite these study limitations, recent guidelines by European Society of Clinical Microbiology and Infectious Diseases (ESCMID) [140] provided a conditional recommendation for the use of target perioperative antibiotic prophylaxis in LT patients colonized by ESBLE, especially in areas with more than 10% of ESBLE prevalence.

#### 3.10.2. Carbapenem-Resistant Enterobacteriaceae

The Center for Disease Control and Prevention (CDC) CRE prevention and control toolkit 2015 recommendations include: healthcare worker education, contact precautions, patient and staff cohorting, chlorhexidine bathing, targeted screening of contacts and active surveillance, optimization of hand hygiene, environmental cleaning, minimized use of indwelling devices, implementation of antimicrobial stewardship, and inter-facility communication [131].

The role of intestinal decolonization and fecal microbiota transplantation for eradication of CRE carriage in SOT recipients is under investigation and remains unclear due to the lack of studies addressing this issue. However, the ESCMID-European Committee on Infection Control (EUCIC) clinical guidelines regarding decolonization of MDR Gram-negative bacteria carriers do not recommend routine decolonization of CRE [141].

When the donor is found colonized or infected with CRE after LT, prompt communication regarding donor cultures and early targeted pre-emptive therapy are essential. On the other hand, if the donor is found to be infected or colonized with CRE prior to transplantation, its suitability for organ donation should be carefully considered, particularly if the infection concerns the liver or is a BSI [70].

A recent review and the European guidelines highlight how the evidence for target prophylaxis for CRE in pre-LT colonized patients is still scarce [140,144]. A recent study suggested the use of risk prediction models for infection and mortality in order to enable better targeting interventions for CRE infection after transplant [93] rather than a target prophylaxis.

#### 3.10.3. MDR, XDR and PDR Non-Fermenting Gram-Negative Bacilli

Contact precautions, improved compliance with hand hygiene, and cohorting are recommended in the settings of MDR *P. aeruginosa*, CRAB, and other non-fermenting Gram negative bacilli [70]. For these organisms, environmental control, deep cleaning, and care of indwelling devices are of particular importance, as they are saprophytic bacteria [133]. These organisms can also colonize devices used for respiratory therapy or diagnostic procedures (e.g., bronchoscopes). Thus, proper disinfection and possibly sterilization are required, and workers using and processing reusable equipment should be properly trained [145].

In addition to the above-mentioned measures, daily chlorhexidine bathing, active rectal and groin screening can be useful infection control bundles to curtail CRAB outbreaks [133,134,135,136].

#### 3.10.4. Vancomycin-Resistant Enterococci

Among infection prevention measures for VRE, the only one found to be beneficial in a recent meta-analysis was hand hygiene after patient encounters [132]. Furthermore, this meta-analysis raised concerns about the quality and limitations of the published infection control studies on VRE.

VRE-targeted LT surgical prophylaxis is not recommended by international guidelines [146]. A study by Sarwar et al. [147] reported that among 27 VRE-colonized patients that underwent LT, 25 received surgical prophylaxis with daptomycin. Notably, none of the 25 patients that received daptomycin developed a VRE infection during the post-transplant period, whilst 2 patients, who did not receive daptomycin, developed VRE bacteraemia.

#### 3.10.5. Methycillin-Resistant *Staphylococcus aureus* (MRSA)

Infection control strategies such as surveillance cultures to detect nasal colonization and decolonization with intranasal mupirocin, the use of cohorting, and contact isolation measures are widely used and related to a decrement of MRSA infection in the general population [148].

Data consistent with the general population are also available for the LT setting. A retrospective study conducted by Singh et al. [114] from 1996 to 2004 showed how the application of these strategies since 2000 reduced the rate of post-LT new acquisition of *S. aureus* nasal colonization from 45.6% to 9.9%. A concomitant decrement of post-LT *S. aureus* infections from 40.4% to 4.1% was also observed, with recipients acquiring *S. aureus* post-LT significatively more at risk of *S. aureus* infection vs. non-carriers and patients colonized before LT. Although this study did not focus specifically on MRSA, these data support the use of the above-mentioned MRSA infection control strategies in LT recipients.

## 4. Summary and Future Perspective

The prevalence of, and risk factors associated with, MDRO colonization and infection in LT candidates/recipients vary widely according to specific pathogens. As MDRO infections profoundly impact LT recipients’ morbidity and mortality, preventive measures are essential. The implementation of infection control and antimicrobial stewardship measures is the keystone for the reduction in MDRO-colonization prevalence. Regarding DDI, the development of standardized definitions of donor-derived MDRO infections are fundamental as centralized reporting mechanisms, with the goal to reduce underreporting of cases. Development and maintenance of updated local and regional guidelines and protocols are necessary to face MDRO infections in LT. Further studies of the epidemiology of LT associated infections, preventive measures of DDI, and the efficacy of a LT surgical antimicrobial prophylaxis tailored to the candidate’s colonization are required.

## Figures and Tables

**Table 1 microorganisms-11-01606-t001:** Risk factors and epidemiology of MDR-colonization and MDR-infections in LT-recipients.

Bacteria	Infection Risk Factors	Pre-LT Colonization Prevalence	Post-LT Infection Prevalence	Post-LT Infection-Related Mortality in Infected Patients
ESBLE [45,46,47]	ESBLE colonization, MELD > 25, reintervention	4–22%	4–27%	15–28% at 30 d
CRE [8,9,14,48,49,50,51,52,53,54,55,56]	CRE colonization, higher MELD at LT, intraoperative blood loss (>1500 mL), prolonged post-LT ICU stay and post-LT intubation, post-LT hemodialysis, combined transplant, biliary complications, reintervention, rejection	3–23%	2–26%	30–70% at 30 d
MDR Pseudomonas aeruginosa [44,57,58,59]	MDR-PA colonization, prolonged post-LT ICU stay and post-LT intubation	2–3%	2–3%	40% at 30 d
CRAB [44,60,61,62]	Pre-LT CRAB colonization, fulminant hepatitis, longer cold ischemia time, prolonged post-LT ICU stay and post-LT intubation, post-LT dialysis	0.3–11%	7–29%	50–65%
VRE [63,64,65,66,67]	VRE colonization, post-LT hemodialysis, length of post-LT hospital stay, bile leak	12–27%	2–9%	NA, but overall mortality increased in VRE-colonized
MRSA [13,58,68,69]	MRSA colonization, decreased prothrombin time ratio	3–13%	11%	6–60% at 30 d

LT: liver transplant, ESBLE: extended-spectrum beta-lactamase producing Enterobacteriaceae, CRE: carbapenem-resistant Enterobacteriaceae, ICU: intensive care unit, MDR: multi-drug-resistant, PA: *Pseudomonas aeruginosa*, CRAB: carbapenem-resistant *Acinetobacter baumannii*, VRE: vancomycin-resistant Enterococci, NA: not available, MRSA: methicillin-resistant *Staphylococcus aureus*, MELD: model for end stage liver disease.

**Table 2 microorganisms-11-01606-t002:** Studies reporting ESBLE colonization, total infections, bloodstream infections, and surgical site infections.

Author, Year	Country	LT Recipients Nr	Pre-LT ESBLE-Colonization	Post-LT ESBLE-Infections	Post-LT ESBLE BSI	Post-LT ESBLE SSI
Bert et al., 2012 [75]	France	710	29 (4.1%)	39 (5.5%)	10 (1.4%)	19 (2.7%)
Kim et al., 2013 [76]	Korea	222	NA	NA	27 (24.1%)	NA
Aguiar et al., 2014 [77]	Brazil	238	NA	NA	17 (7%)	NA
Bert et al., 2014 [47]	France	317	50 (15.7%)	42 (13.2%)	NA	NA
O’Connell et al., 2015 [46]	Ireland	128	28 (21.9%)	NA	NA	NA
Magro et al., 2021 [78]	France	250	47 (19%)	23/190 (12%)	NA	NA
Logre et al., 2021 [45]	France	749	100 (13.3%)	23 (3.5%)	5 (0.7%)	11 (1.5%)

LT: liver transplant, ESBLE: extended spectrum beta-lactamase Enterobacteriaceae, BSI: bloodstream infection, SSI: surgical site infection, NA: not available.

**Table 3 microorganisms-11-01606-t003:** Studies reporting CRE colonization, total infections, bloodstream infections, and surgical site infections.

Author, Year	Country	LT Patients Nr	Pre-LT CRE-Colonization	Post-LT CRE-Infections	Post-LT CRE BSI	Post-LT CRE SSI
Kalpoe et al., 2012 [56]	USA	175	NA	14 (8%)	12 (6.86%)	11 (6.29%)
Lubbert et al., 2014 [88]	Germany	9	2 (22.2%)	NA	5 (55.5%)	NA
Pereira et al., 2015 [55]	USA	304	NA	20 (6.6%)	13 (4.28%)	11 (3.62%)
Mazza et al., 2017 [53]	Italy	310	10 (3.2%)	8 (2.5%)	4 (1.3%)	1 (0.4%)
Freire et al., 2017 [54]	Brazil	386	68 (17.6%)	59 (15.7%)	30 (15.02%)	28 (7.25%)
Macesic et al., 2018 [14]	USA	128	25 (19.5%)	3 (2.3%)	1 (0.78%)	NA
Mularoni et al., 2019 [87]	Italy	526	NA	39 (7.4%)	24 (4.56%)	13 (2.4%)
Errico et al., 2019 [52]	Italy	521	13 (2.5%)	10 (1.9%)	NA	NA
Giannella et al., 2019 [9]	Italy	553	38 (6.8%)	57 (10.3%)	48 (8.6%)	17 (3.07%)
Cinar et al., 2019 [50]	Turkey	142	NA	37 (26%)	NA	NA
Massa et al., 2019 [51]	Greece	44	10 (22.72%)	7 (15.9%)	5 (11.4%)	NA
Chen et al., 2020 [48]	China	387	NA	26 (6.7%)	16 (4.1%)	NA
Freire et al., 2021 [8]	Brazil	762	98 (12.9%)	NA	NA	54 (7.09%)
Schultze et al., 2021 [61]	Germany	351	15 (4.3%)	NA	NA	NA
Taimur et al., 2021 [89]	USA	30	NA	24 (40%)	9 (37.5%)	1 (4.2%)
Liu et al., 2022 [49]	China	272	NA	19 (6.9%)	NA	NA
Freire et al., 2022 [90]	Brazil, Italy	1544	116 (7.5%)	NA	NA	NA

LT: liver transplant, CRE: carbapenem-resistant Enterobacteriaceae, BSI: bloodstream infection, SSI: surgical site infection, NA: not available, USA: United States of America.

**Table 4 microorganisms-11-01606-t004:** Studies reporting MDR-Pseudomonas aeruginosa colonization, total infections, bloodstream infections, and surgical site infections.

Author, Year	Country	LT Patients Nr	Pre-LT MDR-PA Colonization	Post-LT MDR-PA Colonization	Post-LT MDR-PA Infections	Post-LT MDR-PA BSI	Post-LT MDR-PA SSI
Freire et al., 2017 [44]	Brazil	181	5 (2.8%)	22 (12.2%)	6 (3.3%)	NA	NA
Hashimoto et al., 2009 [97]	Japan	170	NA	NA	5 (2.9%)	NA	NA
Zhong et al., 2012 [33]	China	271	NA	NA	6 (2.2%)	NA	NA
Schultze et al., 2021 [61]	Germany	351	7 (2%)	NA	NA	NA	NA

LT: liver transplant, MDR-PA: multi-drug-resistant *Pseudomonas aeruginosa*, BSI: bloodstream infection, SSI: surgical site infection, NA: not available.

**Table 5 microorganisms-11-01606-t005:** Studies reporting CRAB colonization, total infections, bloodstream infections, and surgical site infections.

Author, Year	Country	LT Patients Nr	Pre-LT CRAB Colonization	Post-LT CRAB Colonization	Post-LT CRAB Infections	Post-LT CRAB BSI	Post-LT CRAB SSI
Freire et al., 2016 [98]	Brazil	196	21 (11%)	85 (43%)	56 (29%)	15 (7.6%)	20 (10%)
Zhong et al., 2012 [33]	China	271	NA	NA	20 (7.4%)	NA	NA
Schultze et al., 2021 [61]	Germany	351	1 (0.3%)	NA	NA	NA	NA
Kim et al., 2018 [62]	Korea	393	NA	NA	NA	14 (3.6%)	NA

LT: liver transplant, CRAB: Carabapenem-resistant Acinetobacter baumannii, BSI: bloodstream infection, SSI: surgical site infection, NA: not available.

**Table 6 microorganisms-11-01606-t006:** Studies reporting VRE colonization, total infections, bloodstream infections, and surgical site infections.

Author, Year	Country	LT Patients Nr	Pre-LT VRE Colonization	Post-LT VRE Colonization	Post-LT VRE Infections	Post-LT VRE BSI	Post-LT VRE SSI
Chiang et al., 2022 [66]	Canada	343	68 (19.8%)	27 (9.8%)	6 (2%)	NA	NA
Ejtehadi et al., 2021 [111]	Iran	753	NA	51 (6.8%)	NA	NA	NA
Banach et al., 2016 [67]	USA	61	27 (44%)	NA	3 (5%)	NA	NA
Kim et al., 2015 [65]	South Korea	142	37 (22.8%)	21 (20%)	13 (9%)	0 (0%)	0 (0%)

LT: liver transplant, VRE: vancomycin-resistant enterococcus, BSI: bloodstream infection, SSI: surgical site infection, NA: not available.

**Table 7 microorganisms-11-01606-t007:** Studies reporting MRSA colonization, total infections, bloodstream infections, and surgical site infections.

Author, Year	Country	LT Patients Nr	Pre-LT MRSA Colonization	Post-LT Colonization	MRSA Post-LT Infections	Post-LT MRSA BSI	Post-LT MRSA SSI
Kim et al., 2013 [76]		222	NA	NA	NA	13 (65)	NA
Kim et al., 2015 [65]	South Korea	142	12 (7.4%)	9 (6.9%)	19 (13.4%)	0 (0%)	2 (1.4%)
Takemura et al., 2019 [117]	Japan	106	14 (13%)	NA	NA	42(40%)	NA

LT: liver transplant, MRSA: methicillin-resistant *Stafilococcus aureus*, BSI: bloodstream infection, SSI: surgical site infection, NA: not available.

**Table 8 microorganisms-11-01606-t008:** Studies reporting epidemiological data of DDI in LT-recipients.

Author, Year	Country	LT Patients Nr	DDI in LT	MDRO DDI in LT	Pathogens Involved
Obed et al., 2006 [125]	Germany	1	1	1 (100%)	MRSA
Altman et al., 2014 [124]	USA	1	1	1 (100%)	MRSA
Wendt et al., 2014 [122]	USA	1	1	1 (100%)	MRSA
Miceli et al., 2015 [123]	USA	2	1	1 (50%)	1 MRSA, 1 *E. faecalis*
Mularoni et al., 2015 [128]	Italy	15	NA	2 (1.6%)	CP-*K. pneumoniae*
Errico et al., 2018 [52]	Italy	571	NA	0	-
Xiao et al., 2021 [129]	China	240	NA	6 (2.5%)	2 VRE, 1 CRAB, 2 CP-KP, 1 ESBL-*E. aerogenes*
Anesi et al., 2022 [121]	USA	182	NA	1	-

DDI: donor-derived infections; LT: liver transplant, VRE: vancomycin-resistant *enterococcus*; CP-KP: carbapenem-resistant *K. pneumoniae*; ESBL: extended spectrum beta lactamases, MRSA: methicillin-resistant *S. aureus*, USA: United States of America.

## Data Availability

The data presented in this study are available on request from the corresponding author.

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
