# Peer review of "Epidemiology and Prevention of Early Infections by Multi-Drug-Resistant Organisms in Adults Undergoing Liver Transplant: A Narrative Review"

_microorganisms, 2023, doi:10.3390/microorganisms11061606_

Round 1
Reviewer 1 Report
This is a review on the bacterial infection after liver transplantation.
I have a comment.
1. The organisms can be replaced by bacteria. There is no description on the organisms other than bacteria (virus or fungus) in the text..
The quality of English language is good.
Author Response
The MDRO acronym is used worldwide to identify multidrug resistant bacteria.
Authors think that the usage of the word “organism” is more appropriate than the word “bacteria” in this context.
To be clearer and more precise we change the first sentence in the section 2.4
“MDRO was defined according to the 2012 international consensus[19], as bacteria non-susceptible to at least one agent in three or more antibiotic classes [20].”
And we add this reference
“Centers for Disease Control and Prevention C. National Healthcare Safety Network (NHSN). Patient Safety Component Manual. Multidrug‐Resistant Organism & Clostridioides difficile Infection (MDRO/CDI) Module 2019. https://www.cdc.gov/nhsn/pdfs/ pscmanual/pcsmanual_current.pdf. Accessed May 28, 2023”
Reviewer 2 Report
It is timely for a comprehensive narrative type review to be performed on this topic. However there are some issues with the manuscript in its current form (reflecting perhaps the absence of some of the literature on this topic from North America). The issues identified with the manuscript include-
1) Some of the statements made in the first paragraph of the abstract could do with being revised. Is there enough in the way of published data to support the statements made in the first sentence, if so this needs to be stated. If not then the sentence needs to be rephrased
2) The wording of the second sentence in the first paragraph of the introduction is also confusing and could do with being rewritten. Basically, there are risk factors for organ donors having MDRO and it is increasingly becoming apparent what some of these risk factors are- for eg Anesi JA, Han JH, Lautenbach E, et al. Impact of deceased donor multidrug-resistant bacterial organisms on organ utilization. Am J Transpl 2020; 20: 2559-2566.
3) Section 3.9 on Donor derived MDRO infections needs to be expanded. There is literature out of North America which has not been mentioned for eg the UNOS DTAC reports - https://pubmed.ncbi.nlm.nih.gov/32627325/ as well as other articles via Anesi JA et al (one of which I have already mentioned). This has led to at times guidelines being formulated within regions as to how best to approach the organ donor suspected of having MDRO (of note not all organ donors who are at risk of transmitting MDRO's will have a positive culture available at the time of the organ offer) Donor-derived infections: Guidelines from the American Society of Transplantation Infectious Diseases Community of Practice - PubMed (nih.gov)
4) Section 3.10 also needs work. Some of the generic principles pertaining to the management of liver transplant recipients at risk of developing a donor derived MDRO are scattered throughout this section. This information might be best summarized once and at the beginning of the section, with the specifics related to the organisms of interest then to follow in the sub sections. This prevents redundancy and duplication of the information.
5) Section 4. The Summary and Future Perspective. This seems limited. There is a lot more that needs to be done and could be included in this section. This includes, accurate, centralized reporting mechanisms, standardized definitions of donor derived MDRO, development and maintenance of local and regional guidelines and protocols.
Round 2
Reviewer 2 Report
I think that the authors have undertaken enough in the way of revisions etc of this manuscript for it now to be published